# Striatal cholinergic interneuron pause response requires Kv1 channels, is absent in dyskinetic mice, and is restored by dopamine D5 receptor inverse agonism

Cecilia Tubert[1†], Rodrigo Manuel Paz[1,2†], Agostina Mónica Stahl[1], Kianny Miroslava Sanchez Armijos[1], Lorena Rela[1,3], Mario Gustavo Murer[1,3]*

[1]Universidad de Buenos Aires, CONICET, Instituto de Fisiología y Biofísica Bernardo Houssay, Grupo de Neurociencia de Sistemas, Buenos Aires, Argentina; [2]Department of Neurology, UCSF, San Francisco, United States; [3]Universidad de Buenos Aires, Facultad de Medicina, Departamento de Ciencias Fisiológicas, Buenos Aires, Argentina

*For correspondence:
gmurer@fmed.uba.ar

†These authors contributed equally to this work

Competing interest: The authors declare that no competing interests exist.

## eLife Assessment

The authors investigated the mechanisms underlying the pause in striatal cholinergic interneurons (SCINs) induced by thalamic input, identifying that Kv1 channels play a key role in this burst-dependent pause. The experimental evidence is **convincing**.
The study provides **important** mechanistic insights into how burst activity in SCINs leads to a subsequent pause, highlighting the involvement of D1/D5 receptors.

**Abstract** Striatal cholinergic interneurons (SCINs) exhibit pause responses conveying information about rewarding events, but the mechanisms underlying these pauses remain elusive. Thalamic inputs induce a pause mediated by intrinsic mechanisms and regulated by dopamine D2 receptors (D2Rs), though the underlying membrane currents remain unknown. Moreover, the role of D5 receptors (D5Rs) has not been addressed so far. Here, we performed ex vivo studies showing that glutamate released by thalamic inputs in the dorsolateral striatum induces a burst in SCINs, followed by a pause mediated by the activation of a Kv1-dependent delayed rectifier current. Endogenous dopamine promotes this pause through D2R stimulation, while pharmacological stimulation of D5Rs suppresses it. Remarkably, this pause is absent in parkinsonian mice rendered dyskinetic by chronic L-DOPA treatment but can be reinstated acutely by the inverse D5R agonist clozapine. Blocking the Kv1 current eliminates the pause reinstated by the D5R inverse agonist. In contrast, the D2-type receptor agonists quinpirole and sumanirole failed to reinstate a pause in dyskinetic mice. In conclusion, stimulation of thalamic inputs induces excitation followed by a pause in SCINs, which is lost in parkinsonian mice that have been rendered dyskinetic. This pause is mediated by delayed rectifier Kv1 channels, which are tonically blocked in dyskinetic mice by a mechanism depending on D5R ligand-independent activity. Targeting these alterations may have therapeutic value in Parkinson's disease.

## Introduction

Dopamine (DA), released by neurons from the ventral tegmental area and substantia nigra pars compacta (SNc), and acetylcholine (ACh), mainly released by striatal cholinergic interneurons (SCINs), are the main modulators of cortico-striatal circuits. Both types of neurons are tonically active, and their activity is altered by unexpected rewards or by the sensory cues that predict those rewards. These events evoke a burst in dopaminergic neurons coincident with a pause response in SCINs. The resulting increase in DA and decrease in ACh are critical for striatal-dependent learning and decision-making through modulatory effects on striatal 'medium spiny' projection neurons (MSNs) (*Cragg, 2006*; *Ding et al., 2010*; *Reynolds et al., 2022*).

In vivo recordings show that the phasic response of striatal tonically active neurons (TANs, putative SCINs) to salient events typically involves a brief burst of action potentials followed by a short pause (*Kimura et al., 1984*; *Aosaki et al., 1994a*; *Aosaki et al., 1994b*; *Apicella, 2007*). Recent in vivo studies in rodents show variations in the SCIN pause response across striatal subregions, where the initial excitation preceding the pause is typically seen in the dorsolateral striatum but may be minimal or absent in the dorsomedial region (*Duhne et al., 2024*). The precise mechanisms underlying this response remain elusive. Evidence suggests the involvement of both dopaminergic and thalamic inputs in the generation of the pause. Neurons of the intralaminar nuclei of the thalamus, which send excitatory projections to the striatum, also respond to reward and salient stimuli, and the TAN pause is absent when the thalamus is locally inhibited (*Matsumoto et al., 2001*). Dopamine D2 receptor (D2R) antagonists or genetic elimination of D2Rs from cholinergic neurons have been shown to reduce the duration of the pause (*Watanabe and Kimura, 1998*; *Ding et al., 2010*; *Kharkwal et al., 2016*), implicating a direct contribution of D2R-mediated inhibition of SCINs. Additionally, ex vivo studies reveal heterogeneous responses of SCINs to stimulation of dopaminergic axons, with a more pronounced dopaminergic influence on the pause response in the dorsal vs the ventral striatum (*Chuhma et al., 2014*). Further evidence suggests that the pause in SCINs arises from intrinsic mechanisms activated by depolarization (*Reynolds et al., 2004*; *Wilson and Goldberg, 2006*; *Sanchez et al., 2011*). Experiments conducted in brain slices demonstrate that the pause that follows stimulation of SCIN excitatory inputs persists even after blocking GABA-A receptors, indicating the involvement of voltage (Kv) or calcium-dependent (KCa) potassium currents, with DA likely modulating these currents (*Ding et al., 2010*; *Schulz and Reynolds, 2013*; *Zhang et al., 2018*; *McGuirt et al., 2022*).

The importance of DA is underscored by the absence of pause in animals with chronic nigrostriatal lesions (*Aosaki et al., 1994a*). While reduced D2R stimulation may explain this, genetic elimination of D2Rs from cholinergic neurons reduces but does not eliminate the pause (*Kharkwal et al., 2016*), suggesting that additional factors are involved. SCINs also express dopamine D5 receptors (D5Rs), which have been recently linked to SCIN functional changes in parkinsonian animals (*Paz et al., 2022*). However, their role in regulating phasic SCIN responses remains unaddressed, and whether alterations in D5R signaling contribute to the absence of pause in DA-depleted animals remains unclear.

Here, we report ex vivo studies showing that specific activation of intralaminar thalamostriatal inputs triggers an initial excitation followed by a pause in SCINs recorded from the dorsolateral striatum. This pause requires the activation of an intrinsic delayed rectifier current mediated by Kv1 channels and is modulated by D2Rs and D5Rs. Furthermore, we found that this SCIN pause is absent in parkinsonian mice with chronic L-DOPA treatment, but can be acutely reinstated by the inverse D5R agonist clozapine. Finally, the pause reinstated by clozapine is abolished by blocking the Kv1 current, indicating that inhibition of constitutive D5R signaling releases this current from a tonic restraining effect, aligning with our prior findings (*Paz et al., 2022*).

## Results

### The timing of thalamic input interacts with SCIN intrinsic mechanisms to determine the duration of the pause response

To induce a pause in SCINs in response to thalamic activation, we injected an AAV2-CamKII-ChR2-YFP viral vector into the intralaminar nuclei of the thalamus in adult Chat[Cre/+];Rosa26[LSL-tdTomato/+] mice. We used these mice to facilitate the localization of SCINs in brain slices (*Tubert et al., 2016*). Four weeks later, we sacrificed the animals and made coronal slices of the striatum for ex vivo patch clamp recordings (*Figure 1A*). Histological analysis confirmed the site of AAV injection and the labeling of

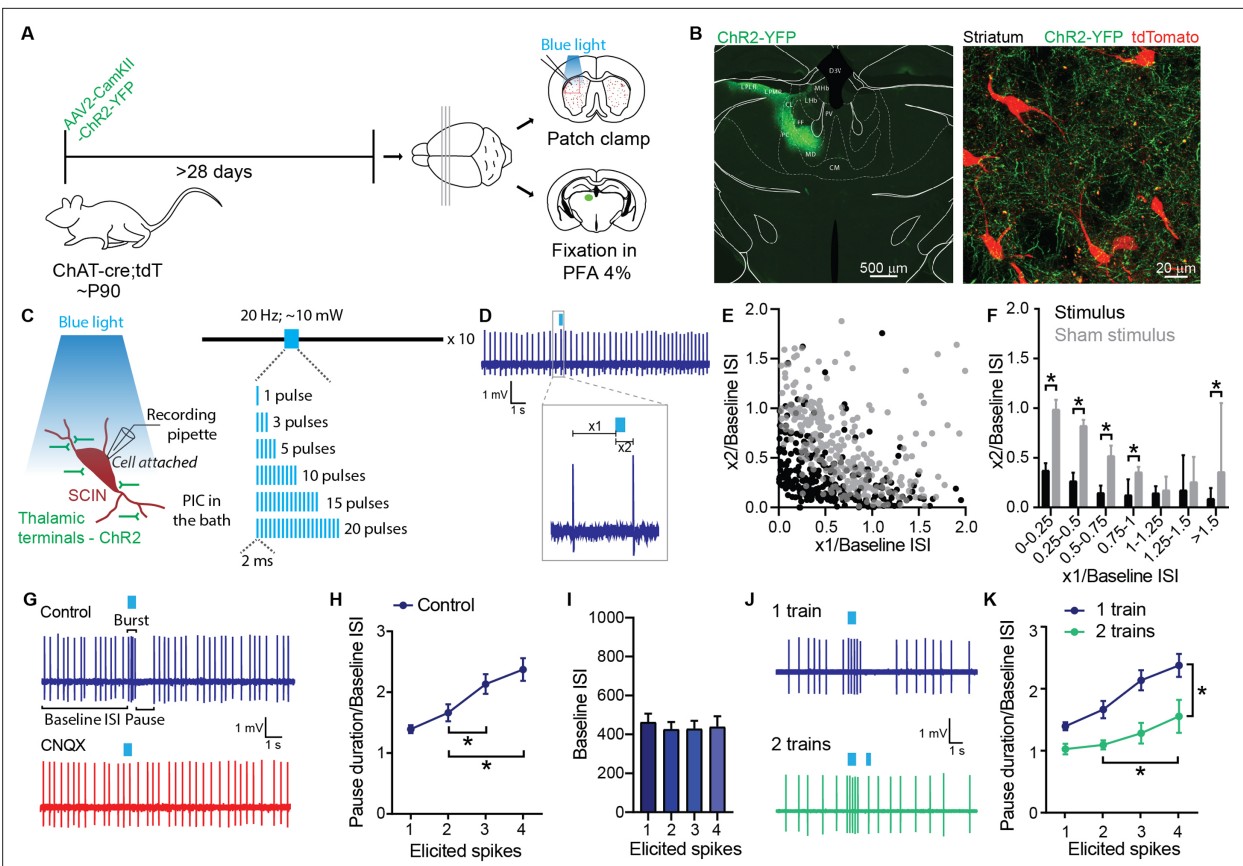

**Figure 1.** Striatal cholinergic interneurons (SCIN) burst-pause response to activation of thalamic terminals. (**A**) Experimental design. (**B**) Micrograph showing the thalamic region transfected with ChR2-YFP (left) and a striatal region (z-stack maximal projection) showing tdTomato-reported SCINs and thalamic terminals expressing ChR2-YFP (right). (**C**) Schematic representation of optogenetic stimulation of thalamic inputs to a SCIN recorded in cell-attached configuration. Picrotoxin (PIC, 100 µM; GABA-A antagonist) was present in the bath in all experiments. (**D**) Representative recording of a SCIN that responded to optogenetic activation of thalamic terminals with 1 spike. Inset: Details of the measurements of x1 and x2 values. x1: time from the last spike before the optogenetic stimulation and the start of the stimulation; x2: time from the initiation of stimulation to the following spike. (**E**) x2/baseline ISI vs x1/baseline ISI for single spike responses to optogenetic activation of thalamic terminals (black dots) and for sham stimulation (gray dots). (**F**) Data shown in E were binned and expressed as median and 95% confidence interval. *p<0.05, Mann-Whitney U test. (**G**) Representative recordings of the pause response of a SCIN to optogenetic stimulation of thalamic terminals in cell-attached configuration before and after bath application of CNQX (20 µM; AMPA receptor antagonist). (**H**) Pause duration/baseline ISI of SCINs that responded with 1, 2, 3, or 4 spikes to optogenetic activation of thalamic terminals (one-way RM ANOVA, treatment 2 vs 3 spikes: *p=0.0032; 2 vs 4 spikes: *p=0.0017). (**I**) Baseline ISI of SCINs that responded with 1, 2, 3, or 4 spikes to optogenetic activation of thalamic terminals (one-way RM ANOVA, ns). (**J**) Representative recordings of SCIN stimulated with one train of optogenetic activation of thalamic terminals (top) or with two trains (bottom). (**K**) Pause duration/baseline ISI of SCINs that responded with 1, 2, 3, or 4 spikes to optogenetic activation of thalamic terminals, with one or two trains of stimulation pulses. The second train occurs 350 ms after the first train (two-way RM ANOVA, interaction ns; treatment *p<0.0001; elicited spikes *p<0.0001; 2 vs 4 spikes *p=0.0042). Mean ± SEM; n=11–18 cells per group, from >5 mice.

The online version of this article includes the following source data for figure 1:

**Source data 1.** Striatal cholinergic interneurons (SCIN) burst-pause response to activation of thalamic terminals.

thalamostriatal projections (*Figure 1B*). We recorded SCINs in the cell-attached configuration and stimulated thalamic axons with 2-ms-long pulses of blue light at 20 Hz and 10 mW (*Figure 1C*). All the experiments were performed in the presence of picrotoxin (PIC, 100 µM) in the bath. When the stimulation elicited a minimal response (1 spike), the latency of the evoked spike decreased with the time separating the stimulus from the preceding spontaneous spike (*Figure 1D and E*). Thus, a stimulus arriving after an average baseline interspike interval (ISI) has elapsed produces a shorter latency response than a stimulus arriving shortly after the preceding spontaneous spike (*Figure 1E*), as expected given the strong after-hyperpolarization (AHP) that follows SCIN spikes (*Reynolds et al., 2004*; *Wilson and Goldberg, 2006*). While a similar inverse relationship is observed under conditions

of 'sham stimulation' (**Figure 1E**), after optical stimulation, the response latencies were significantly smaller at intervals shorter and longer than the average ISI, indicating that the stimulation increased spike probability (**Figure 1F**). Remarkably, the interval that followed these minimal responses was 30% longer than the average ISI (baseline ISI: 464.61±42.44 ms; pause after 1 elicited spike: 609.3±59.5; p=0.0181, paired t-test), indicating that even an initial excitatory response consisting in a single spike delays the next SCIN spike.

By increasing the number of pulses in the stimulation train, we obtained responses of 2–4 spikes ('bursts') followed by an interruption of SCIN firing, which were abolished by application of an AMPA receptor antagonist in the bath (normalized firing rate (burst/baseline): PIC = 3.611; PIC+CNQX = 1.343; paired t-test, p=0.0409; n=5 cells from 5 animals) (**Figure 1G**). Different train durations were required to evoke a given burst response across SCINs, probably relating to variability in the density of ChR2-expressing afferent fibers across experiments. To minimize the effect of this source of variability, we studied the pause duration as a function of the number of spikes in the burst evoked by thalamic stimulation. The duration of the SCIN pause increased with the number of spikes in the burst (**Figure 1H**), without changes in baseline ISI (**Figure 1I**), as expected if the pause depended on the recruitment of a depolarization-activated hyperpolarizing current (average pause durations: 1 elicited spike = 609.3±59.5 ms; 2 elicited spikes = 744.2±92.4 ms; 3 elicited spikes = 964.7±92.3 ms; 4 elicited spikes = 1065.7±136.5 ms; baseline ISI: 1 elicited spike = 464.6±42.4 ms; 2 elicited spikes = 427.7±36.3 ms; 3 elicited spikes = 430.5±39.9 ms; 4 elicited spikes = 440.3±53.4 ms).

A pioneering in vivo study showed that the thalamic neurons presumptively driving the burst-pause response of SCINs show a burst followed by a pause that ends with a rebound of activity, in connection with reward-related events. This thalamic burst-pause response temporally coincides with the burst-pause response of the presumptive SCINs (**Matsumoto et al., 2001**). This suggests that a decrease in excitatory afferent activity may be necessary for the expression of the SCIN pause. To evaluate the effect of a second volley of excitatory inputs on the SCIN pause duration, we delivered a second thalamic stimulus train 350 ms after the end of the first stimulus. We found that the second stimulus train could terminate the pause, although its effect is delayed as more spikes are recruited in the burst by the first stimulus (**Figure 1J and K**). This is consistent with a graded recruitment of an intrinsic current that reduces the excitability of SCINs, delaying their response to the second stimulus.

## The pause response to thalamic stimulation requires activation of Kv1 channels

In previous work, we found that Kv delayed rectifier channels constituted by Kv1.3 and Kv1.1 subunits contribute to the slow AHP current in SCINs of adult mice (**Tubert et al., 2016**). Moreover, a train of EPSPs evoked by thalamic afferents stimulation is followed by an AHP, which persists in the presence of the GABA-A receptor antagonist PIC and is blocked by the Kv1.3 channel blocker margatoxin (MgTx) (**Tubert et al., 2016**). We asked whether this Kv1 current is recruited during the SCIN pause response. Both MgTx at 30 nM – but not at 3 nM – and dendrotoxin (DTx, 100 nM, a Kv1.1 and Kv1.6 channel blocker) abolished the pause response without changing the spontaneous firing rate or the burst duration (**Figure 2A–E**). Previous studies showed that Kv1.1 and Kv1.3 mRNAs are expressed in SCINs but not in striatal tissue samples lacking SCINs (**Tubert et al., 2016**), consistent with independent findings showing that Kv1.3 mRNA is absent in MSNs (**Shen et al., 2004**). These findings suggest that the effect of MgTx on the pause response is likely mediated by direct actions on SCINs.

SCINs have a high reserve of Kv7 channels that limit EPSP summation but do not seem to contribute to the slow AHP (**Paz et al., 2018**) or pause response (**McGuirt et al., 2022**) in adult mice. Consistent with these findings, the pause induced by thalamic stimulation was not reduced by the Kv7 channel blocker XE-991 (10 μM) (**Figure 2F–I**). Moreover, SCIN hyperpolarizations can be amplified by the recruitment of inwardly rectifying Kir2 channels, presumptively of the Kir2.2 subtype (**Paz et al., 2021**), which could then contribute to the pause response (**Figure 2A**). However, a low concentration of barium that preferentially blocks Kir2.2 channels (10 μM) had no effect on firing frequency, burst duration, or pause response of SCINs to thalamic stimulation (**Figure 2F–I**).

Overall, the data show that the Kv1 current is necessary for the expression of the pause that follows the stimulation of thalamic inputs in SCINs of adult mice.

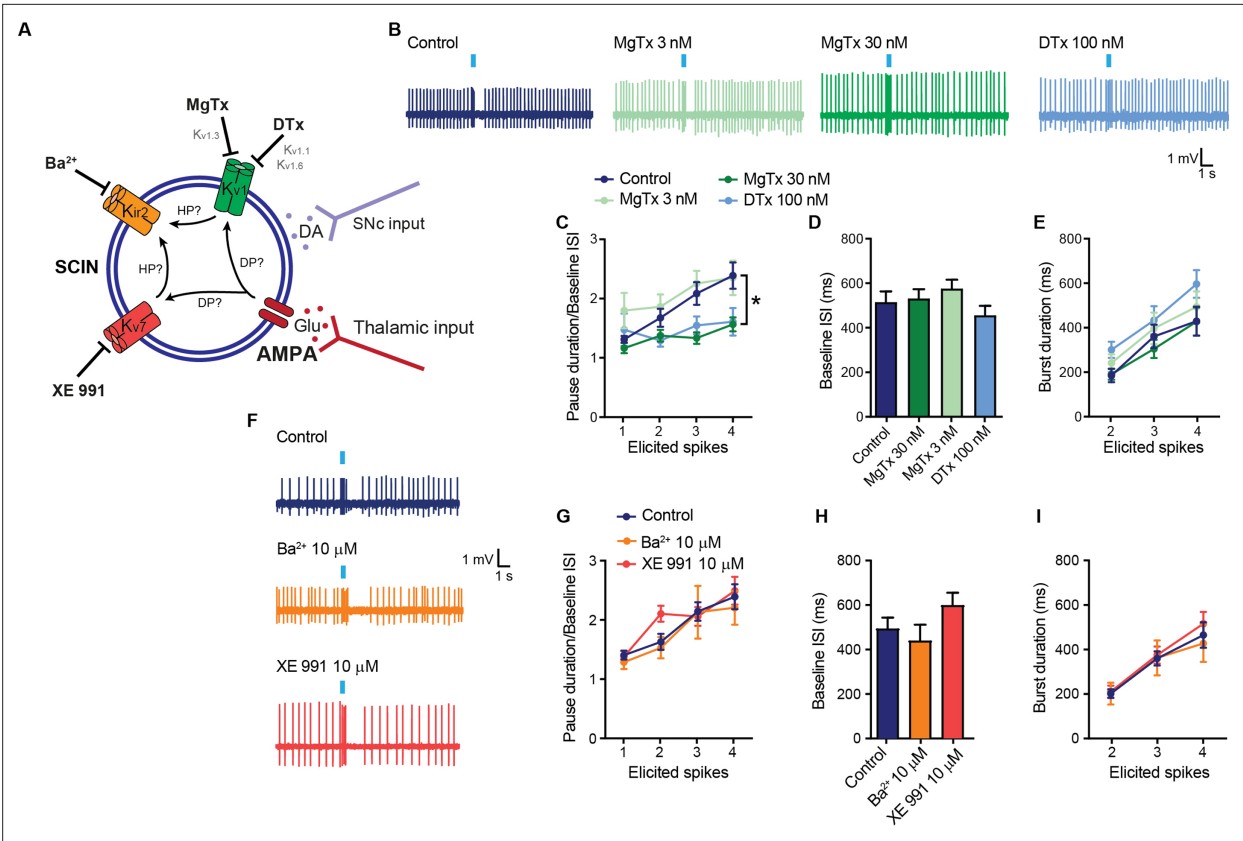

**Figure 2.** Contribution of the Kv1 current to the pause response. (**A**) Schematic representation of the tested hypotheses. DP: depolarization; HP: hyperpolarization; Glu: glutamate; DA: dopamine. (**B**) Representative recordings of striatal cholinergic interneurons (SCINs) in response to optogenetic activation of thalamic terminals, with or without margatoxin (MgTx; Kv1.3 channel blocker) or dendrotoxin (DTx; Kv1.1 and Kv1.6 channels blocker) in the bath. (**C**) Pause duration/baseline ISI of SCINs that responded with 1, 2, 3, or 4 spikes to optogenetic activation of thalamic terminals, with or without MgTx 3 nM, 30 nM, or DTx 100 nM, in the bath (two-way RM ANOVA interaction ns, treatment *p=0.0051; control-MgTx 30 nM: *p=0.0005; control-MgTx 3 nM: ns; control-DTX: *p=0.0462; MgTx 30 nM-MgTx 3 nM: *p<0.0001; MgTx 30 nM-DTX: ns; MgTx 3 nM-DTX: *p=0.0027). (**D**) Baseline ISI of SCINs recorded with or without MgTx 3 nM, 30 nM, or DTx (one-way ANOVA, ns). (**E**) Burst duration of SCINs that responded with 2, 3, or 4 spikes to optogenetic activation of thalamic terminals, with or without MgTx 3 nM, 30 nM, or DTx 100 nM, in the bath (two-way RM ANOVA, ns). (**F**) Representative recordings of SCINs in response to optogenetic activation of thalamic terminals, with or without Ba$^{2+}$ (10 µM; Kir2.2 channel blocker at this concentration) or XE 991 (10 µM; Kv7 channel blocker) in the bath. (**G**) Pause duration/baseline ISI of SCINs that responded with 1, 2, 3, or 4 spikes to optogenetic activation of thalamic terminals, with or without 10 µM XE 991 or 10 µM Ba$^{2+}$ (two-way RM ANOVA, ns). (**H**) Baseline ISI of SCINs recorded with or without Ba$^{2+}$ or XE 991 (one-way ANOVA, ns). (**I**) Burst duration of SCINs that responded with 2, 3, or 4 spikes to optogenetic activation of thalamic terminals, for the above conditions (two-way RM ANOVA, ns). Mean ± SEM; n=7–16 cells per group, from >5 mice.

The online version of this article includes the following source data for figure 2:

**Source data 1.** Contribution of the Kv1 current to the pause response.

## Activation of D5Rs reduces the pause generated in response to thalamic stimulation

Previous in vivo and ex vivo studies have shown that DA promotes the pause response through D2R-mediated actions (**Aosaki et al., 1994a**; **Watanabe and Kimura, 1998**; **Deng et al., 2007**; **Ding et al., 2010**), including inhibitory effects on Ih, which could be activated by the pause and contribute to its end (**Deng et al., 2007**; **McGuirt et al., 2022**). The pause shortens when D2R endogenous activation is blocked with D2-type receptor antagonists (**Aosaki et al., 1994a**; **Watanabe and Kimura, 1998**; **Ding et al., 2010**) or D2Rs are genetically eliminated (**Kharkwal et al., 2016**), and lengthens when D2Rs are overexpressed in SCINs (**Gallo et al., 2022**). In addition to D2Rs, SCINs express D5Rs that increase SCIN excitability by reducing the Kv1-mediated delayed rectifier current (**Paz et al., 2021**; **Paz et al., 2022**), shown above to be involved in the pause response. To assess whether D5R modulates the pause response induced by thalamic stimulation, we tested the effect of the

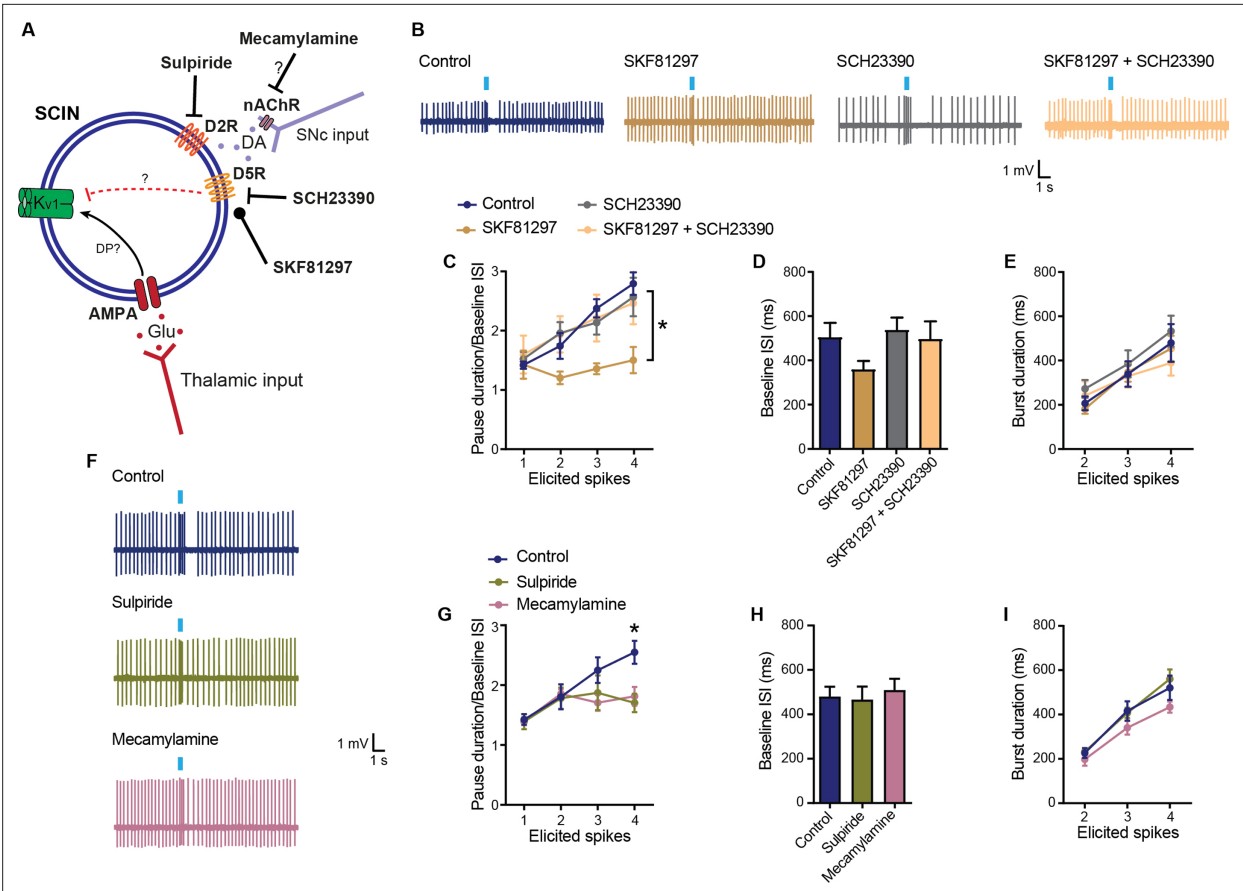

**Figure 3.** Modulation of the pause response by dopamine receptors. (**A**) Schematic representation of the tested hypotheses. DP: depolarization. (**B**) Representative responses of striatal cholinergic interneurons (SCINs) to optogenetic activation of thalamic terminals, with SKF81297 (2 µM; D1/ D5 selective agonist), SCH23390 (10 µM; D1/D5 selective antagonist), or SKF81297+SCH23390 in the bath. (**C**) Pause duration/baseline ISI of SCINs that responded with 1, 2, 3, or 4 spikes to optogenetic activation of thalamic terminals, under the above conditions (two-way RM ANOVA, interaction p=0.0381, Dunnett's multiple comparisons test: control vs SKF81297: 3 spikes *p=0.0001; 4 spikes *p=0.0010.). (**D**) Baseline ISI of SCINs that responded with 1, 2, 3, or 4 spikes to optogenetic activation of thalamic terminals, under the above conditions (one-way ANOVA, *p=0.0906). (**E**) Burst duration of SCINs that responded with 2, 3, or 4 spikes to optogenetic activation of thalamic terminals, under the above conditions (two-way RM ANOVA, ns). (**F**) Representative responses of SCINs to optogenetic activation of thalamic terminals, with sulpiride (10 µM; D2-type receptor selective antagonist) or mecamylamine (10 µM; nicotinic receptor antagonist) in the bath. (**G**) Pause duration/baseline ISI of SCINs that responded with 1, 2, 3, or 4 spikes to optogenetic activation of thalamic terminals, under the above conditions (two-way RM ANOVA, interaction p=0.0369, Dunnett's multiple comparisons test: control vs sulpiride: 4 spikes *p=0.0056; control vs mecamylamine: 4 spikes *p=0.0167.). (**H**) Baseline ISI of SCIN that responded with 1, 2, 3, or 4 spikes to optogenetic activation of thalamic terminals, under the above conditions (one-way ANOVA, ns). (**I**) Burst duration of SCINs that responded with 2, 3, or 4 spikes to optogenetic activation of thalamic terminals, under the above conditions (two-way RM ANOVA, ns). Mean ± SEM; n=7–15 cells per group, from >5 mice.

The online version of this article includes the following source data for figure 3:

**Source data 1.** Modulation of the pause response by dopamine receptors.

D1/D5R selective agonist SKF81297 in our ex vivo preparation. SKF81297 suppressed this pause without changing SCIN spontaneous firing rate, although there was a trend toward shorter baseline ISIs under SKF81297 alone (***Figure 3A–E***). The effect of SKF81297 was prevented by co-application of the selective D1/D5R antagonist SCH23390, which had no effects by itself, ruling out tonic effects of DA through D5Rs (***Figure 3A–E***). Furthermore, we found a small effect of the selective D2R antagonist sulpiride, limited to the pauses that followed 4-spike bursts, supporting a modulatory role of D2Rs (***Figure 3F–I***). Previous studies suggested that synchronized SCIN bursts induce DA release through local effects mediated by nicotinic receptors located in dopaminergic terminals (***Threlfell et al., 2012***). Accordingly, the nicotinic receptor antagonist mecamylamine also reduced the pause that followed 4-spike bursts (***Figure 3F–I***). Presumably, endogenous levels of DA in the slice, produced by

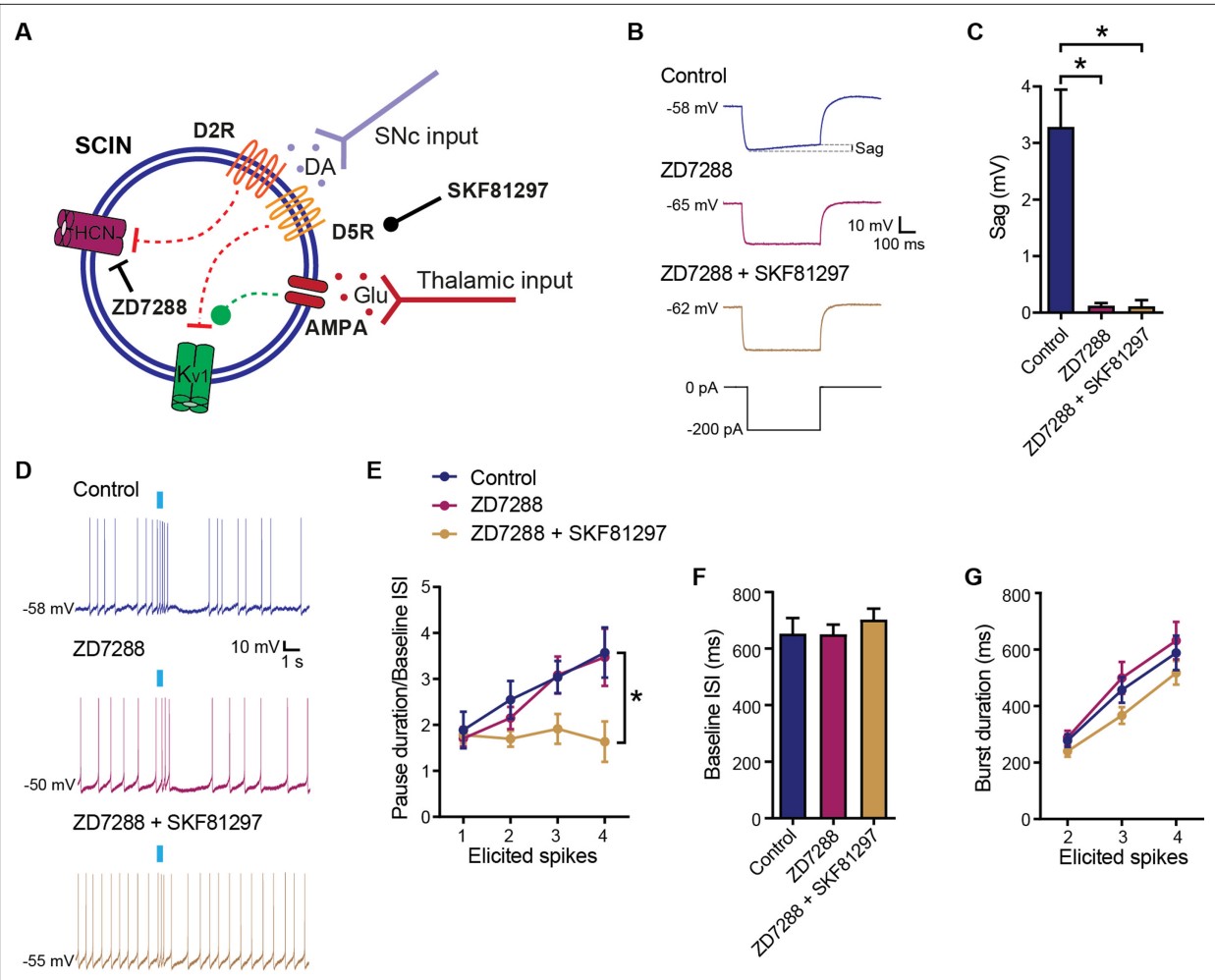

**Figure 4.** D5 receptor (D5R) modulation independent of Ih current. (**A**) Schematic representation of the tested hypotheses. (**B**) Time matched representative responses of striatal cholinergic interneurons (SCINs) to hyperpolarizing current steps with or without ZD7288 (30 μM; HCN channel blocker) in the recording pipette, and with or without SKF81297 (2 μM; D1/D5 selective agonist) in the bath. (**C**) Sag evaluated in the hyperpolarizing step, under the above conditions (one-way ANOVA), p<0.0001; Tukey's multiple comparisons test: control vs ZD7288 *p<0.0001; control vs ZD7288+SKF81297 *p<0.0001. (**D**) Representative responses in whole-cell configuration to optogenetic activation of thalamic terminals, under the above conditions. (**E**) Pause duration/baseline ISI of SCINs that responded with 1, 2, 3, or 4 spikes to optogenetic activation of thalamic terminals, under the above conditions (two-way RM ANOVA, interaction ns, treatment p=0.0373; control vs ZD7288+SKF81297: *p=0.0005; ZD7288 vs ZD7288+SKF81297: *p=0.0067). (**F**) Baseline ISI of SCINs that responded with 1, 2, 3, or 4 spikes to optogenetic activation of thalamic terminals, under the above conditions (one-way ANOVA, ns). (**G**) Burst duration of SCINs that responded with 2, 3, or 4 spikes to optogenetic activation of thalamic terminals, under the above conditions (two-way RM ANOVA, ns). Mean ± SEM; n=13/14 cells per group, from >5 mice.

The online version of this article includes the following source data for figure 4:

**Source data 1.** D5 receptor (D5R) modulation independent of Ih current.

tonic release mechanisms or phasic cholinergic activation of dopaminergic terminals, are sufficient to stimulate D2Rs but not D5Rs.

## Reduction of Ih current does not block the pause generated in response to thalamic activation

To evaluate if D5R stimulation reduces the pause response in our ex vivo preparation through effects on Ih current, we assessed if the SKF81297 effect persisted in the presence of the HCN channel blocker ZD7288 (*Figure 4A*). We performed whole-cell patch clamp recordings, including ZD7288 in the recording pipette (*Paz et al., 2021*). ZD7288 blocked HCN channels as shown by the disappearance of the sag induced by hyperpolarizing steps (*Figure 4B and C*). Because HCN channel block also

stopped spontaneous activity, we injected current through the electrode to induce spiking. In these conditions, thalamic input stimulation induced a burst-pause response. Additionally, the suppression of the pause by stimulation of D5Rs with SKF81297 was not prevented by Ih block (*Figure 4D–G*).

Our results, so far, extend previous work by showing that D5R can inhibit the pause that follows excitation induced by thalamic afferents through effects that are independent of GABA-A receptors, burst amplitude, and Ih current, likely involving Kv1 current inhibition.

## SCINs from L-DOPA-treated parkinsonian mice show reduced pause responses to thalamic activation

Previous in vivo work has shown that the pause response in putative SCINs is absent in an animal model of Parkinson's disease and is partially restored by acute administration of the DA receptor

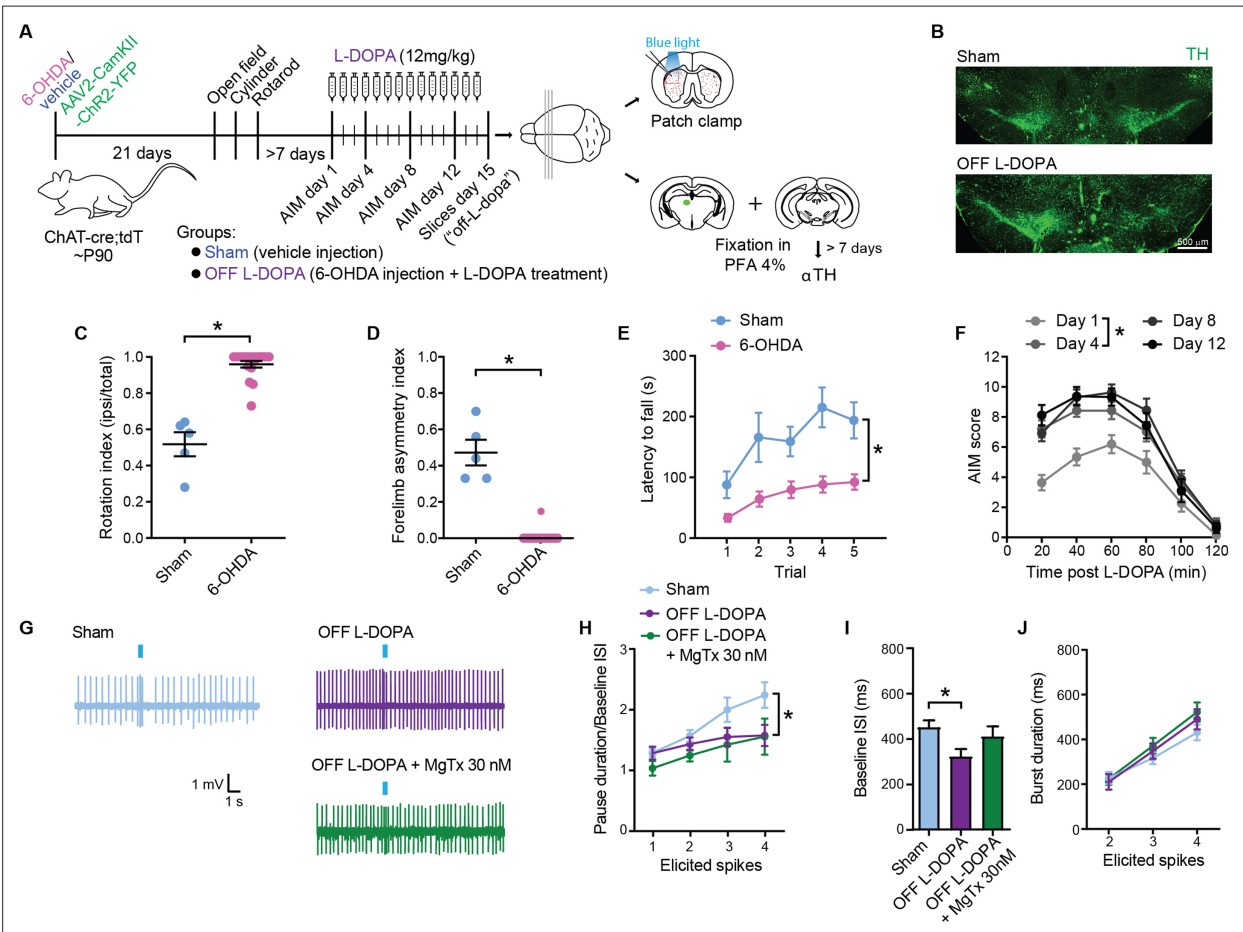

**Figure 5.** Striatal cholinergic interneurons (SCINs) from L-DOPA-treated parkinsonian mice show a reduced pause response. (**A**) Experimental design. (**B**) Micrographs showing TH immunostaining at the level of the substantia nigra. (**C–E**) Rotation index (C), forelimb asymmetry (D), and latency to fall from the rotarod (E) for sham and 6-OHDA mice (C–D: unpaired t test, **p<0.0001; E two-way RM ANOVA, interaction: *p=0.0284, n=5 sham and 22 6-OHDA mice). (**F**) Abnormal involuntary movement (AIM) score of chronically L-DOPA-treated 6-OHDA mice (two-way RM ANOVA, interaction: p=0.0081; post hoc: day 1 vs day 4 *p<0.03). (**G**) Representative cell-attached recordings of the pause response of SCINs to optogenetic stimulation of thalamic terminals, in a sham and a dyskinetic mouse in the OFF L-DOPA condition, with or without margatoxin (MgTx) 30 nM in the bath. (**H**) Pause duration/baseline ISI of SCINs that responded with 1, 2, 3, or 4 spikes to optogenetic activation of thalamic terminals (two-way RM ANOVA, interaction ns, treatment *p=0.0227; Sham vs OFF L-DOPA: *p=0.0301; OFF L-DOPA vs OFF L-DOPA+MgTx: ns; Sham vs OFF L-DOPA+MgTx: *p=0.0032). (**I**) Baseline ISI of SCIN that responded with 1, 2, 3, or 4 spikes to optogenetic activation of thalamic terminals under the above conditions (one-way ANOVA, *p=0.0254; Sham vs OFF L-DOPA: *p=0.0201; OFF L-DOPA vs OFF L-DOPA+MgTx: ns; Sham vs OFF L-DOPA+MgTx: ns). (**J**) Burst duration of SCINs that responded with 2, 3, or 4 spikes to optogenetic activation of thalamic terminals under the above conditions (two-way RM ANOVA, ns). Mean ± SEM; n=14–27 cells per group, from >5 mice.

The online version of this article includes the following source data for figure 5:

**Source data 1.** Striatal cholinergic interneurons (SCINs) from L-DOPA-treated parkinsonian mice show a reduced pause response.

agonist apomorphine (*Aosaki et al., 1994a*). Chronic administration of L-DOPA to parkinsonian animals produces lasting changes in SCIN membrane excitability involving Kv1, Kir2, and likely, other currents (*McKinley et al., 2019*; *Choi et al., 2020*; *Paz et al., 2021*; *Paz et al., 2022*). Yet, whether chronic L-DOPA modifies the SCIN pause response has not been addressed before. In our brain slice preparation, we studied the pause response induced by thalamic afferent stimulation in SCINs from parkinsonian mice treated for 14 consecutive days with L-DOPA (12 mg/kg) or vehicle (*Figure 5A*). Mice unilaterally lesioned with 6-OHDA showed extensive loss of TH-positive neurons in SNc (*Figure 5B*), accompanied by marked parkinsonian-like motor impairment (*Figure 5C–E*). Moreover, these mice developed dyskinesia in response to L-DOPA administration, characterized by an abnormal involuntary movement (AIM) score that worsens after day 4 of treatment (*Figure 5F*). In previous studies, we have shown that the Kv1 current is diminished after lesioning nigral dopaminergic neurons and further suppressed when recorded 24 hr after the end of a chronic L-DOPA treatment (*Paz et al., 2022*). Cell-attached recordings performed 24 hr after the last L-DOPA challenge (OFF L-DOPA condition) showed that the pause is absent in SCINs from dyskinetic mice (*Figure 5G and H*), regardless of the burst duration (*Figure 5J*). Consistent with previous findings (*Paz et al., 2021*), the spontaneous firing rate was higher in SCINs recorded from mice in the OFF L-DOPA condition compared with sham mice (*Figure 5I*). In addition, the inclusion of MgTx in the bath did not have any effect on pause duration, baseline firing rate, and burst duration (*Figure 5G–J*), as expected following previous findings, showing a lack of effect of MgTx on membrane excitability in the same animal model (*Paz et al., 2022*).

These results support that a reduction of the Kv1 current is involved in the loss of the pause response in dyskinetic mice.

## Stimulation of D2Rs in L-DOPA-treated parkinsonian mice does not restore the SCIN pause response

Because the pause response in SCINs appears to partially depend on endogenous D2R activation in DA-intact mice, we tested whether exogenous stimulation of these receptors could restore the pause response in dyskinetic mice in the OFF L-DOPA condition. Contrary to expectations, D2R activation failed to restore the pause response induced by optogenetic stimulation of thalamic afferents, either with quinpirole (1 µM) or with the selective D2 agonist sumanirole (10 µM) (*Figure 6A and B*). Higher concentrations of quinpirole (10 µM) were also ineffective (baseline ISI: OFF L-DOPA=300.8 ± 40 ms,

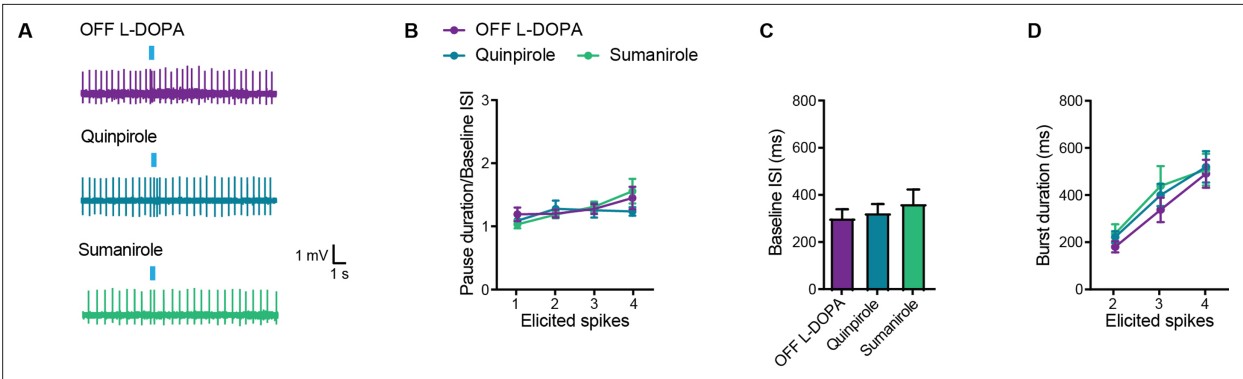

**Figure 6.** D2-type receptor agonists do not restore the pause response in striatal cholinergic interneurons (SCINs) from OFF L-DOPA parkinsonian mice. (**A**) Representative cell-attached recordings of the pause response of SCINs to optogenetic stimulation of thalamic terminals, in a dyskinetic mouse in the OFF L-DOPA condition, with or without quinpirole 1 µM or sumanirole 10 µM in the bath. (**B**) Pause duration/baseline interspike interval (ISI) of SCINs from OFF L-DOPA dyskinetic mice that responded with 1, 2, 3, or 4 spikes to optogenetic activation of thalamic terminals (two-way RM ANOVA, interaction ns, treatment ns). (**C**) Baseline ISI of SCINs that responded with 1, 2, 3, or 4 spikes to optogenetic activation of thalamic terminals, under the above conditions (one-way ANOVA, ns). (**D**) Burst duration of SCINs from dyskinetic mice OFF L-DOPA that responded with 2, 3, or 4 spikes to optogenetic activation of thalamic terminals, under the above conditions (two-way RM ANOVA, interaction ns, treatment ns). Mean ± SEM; n=10–14 cells per group, from >5 mice.

The online version of this article includes the following source data for figure 6:

**Source data 1.** D2-type receptor agonists do not restore the pause response in striatal cholinergic interneurons (SCINs) from OFF L-DOPA parkinsonian mice.

quinpirole = 285.6 ± 40 ms, NS; pause duration/baseline ISI: OFF L-DOPA=1.46 ± 0.18, quinpirole = 1.39 ± 0.16, NS; 14 cells from 5 mice and 3 cells from 3 mice, respectively). In addition, D2R stimulation had no effect on the spontaneous firing rate of SCINs (*Figure 6C*) or on the duration of burst responses evoked by thalamic input (*Figure 6D*).

These findings indicate that D2R stimulation alone is insufficient to restore the SCIN pause in dyskinetic mice.

## Reducing D5R ligand-independent activity restores the SCIN pause response

In a previous study, we found that the physiological changes observed in SCINs of dyskinetic mice depend on increased ligand-independent activity of D5Rs (*Paz et al., 2022*). *Paz et al., 2022*, recorded SCINs in the parkinsonian OFF-L-DOPA condition and showed an increase of membrane excitability that mimics changes acutely induced by SKF81297 in control mice. This hyperexcitability phenotype was reversed by acute administration of drugs that reduce D5R ligand-independent activity (i.e. inverse agonists such as clozapine or flupentixol), as well as by acutely reducing intracellular cAMP signaling (*Paz et al., 2022*). Interestingly, the hyperexcitability of SCINs recorded in the OFF L-DOPA condition was not further enhanced by SKF81297 or MgTx (*Paz et al., 2022*), suggesting D5R-cAMP-Kv1 inhibitory signaling is already working at full steam in these cells. Based on these findings, we speculated that, if Kv1 current suppression underlies the loss of the SCIN pause response in our slice preparation, then D1/D5 inverse agonists should be able to restore it.

The atypical antipsychotic clozapine behaves as an antagonist of D1/D5 and other DA receptors when an agonist is present, but has inverse agonist effects on D5Rs in the absence of an agonist (*Zhang et al., 2014*). Remarkably, clozapine restores the pause in SCINs from dyskinetic mice without changing their firing rate and regardless of the burst duration (*Figure 7A–E*).

To assess whether this effect was mediated through D1/D5Rs, we co-applied the selective D1/D5 antagonist SCH23390. We found that SCH23390 alone had no effect in the SCIN response to thalamic stimulation, as expected given its lack of inverse agonist effects (*Tiberi and Caron, 1994*) and the DA-depleted condition of the slice preparation. However, SCH23390 prevented the pause-restoring effect of clozapine (*Figure 7A–E*), indicating that engagement of D1/D5Rs is required.

Because clozapine also binds to serotonin receptors (5-HTR), we repeated the experiments in the presence of the broad-spectrum 5-HTR antagonist methiothepin. Methiothepin did not block the effect of clozapine (*Figure 7A–E*), arguing against a serotonergic mechanism.

Finally, to confirm that the pause restored by clozapine involves Kv1 channels, we co-applied clozapine and MgTx to the bath (*Figure 7A*). MgTx occluded the pause-restoring effect of clozapine (*Figure 7B–E*), indicating that reducing D5R ligand-independent activity reinstates the Kv1 current in SCINs from OFF L-DOPA parkinsonian mice.

## Discussion

While previous studies support the role of intrinsic membrane currents in the pause response of SCINs, the specific currents involved remain debated. Theories implicating deactivation of Ih current as its primary cause were refuted by experiments demonstrating pause induction after Ih current blockade provided the resulting hyperpolarization is compensated by current injection (*Zhang et al., 2018*; present findings). Although Ih likely contributes to a rebound depolarization limiting pause duration (*Deng et al., 2007*; *McGuirt et al., 2022*), it does not appear to be essential for pause initiation. Further work suggested the involvement of a slow current mediated by Kv7 channels in juvenile mice (*Zhang et al., 2018*), but this current is developmentally downregulated and does not seem to contribute significantly in adult SCINs (*McGuirt et al., 2022*; present findings), even though SCINs retain a high reserve of Kv7 channels that remain responsive to pharmacological activators (*Paz et al., 2018*). Whether this Kv7 channel reserve is recruited under particular conditions not captured in our slice preparation remains to be determined.

Previously, we found that SCIN excitability is strongly modulated by Kv1 channels (*Tubert et al., 2016*). Heteromeric channels constituted by Kv1.3 and Kv1.1 subunits can be blocked by the binding of a single MgTx molecule (*Hopkins, 1998*). In SCINs, MgTx enhances subthreshold EPSP summation and blocks the subsequent AHP (*Tubert et al., 2016*). The present study extends these findings

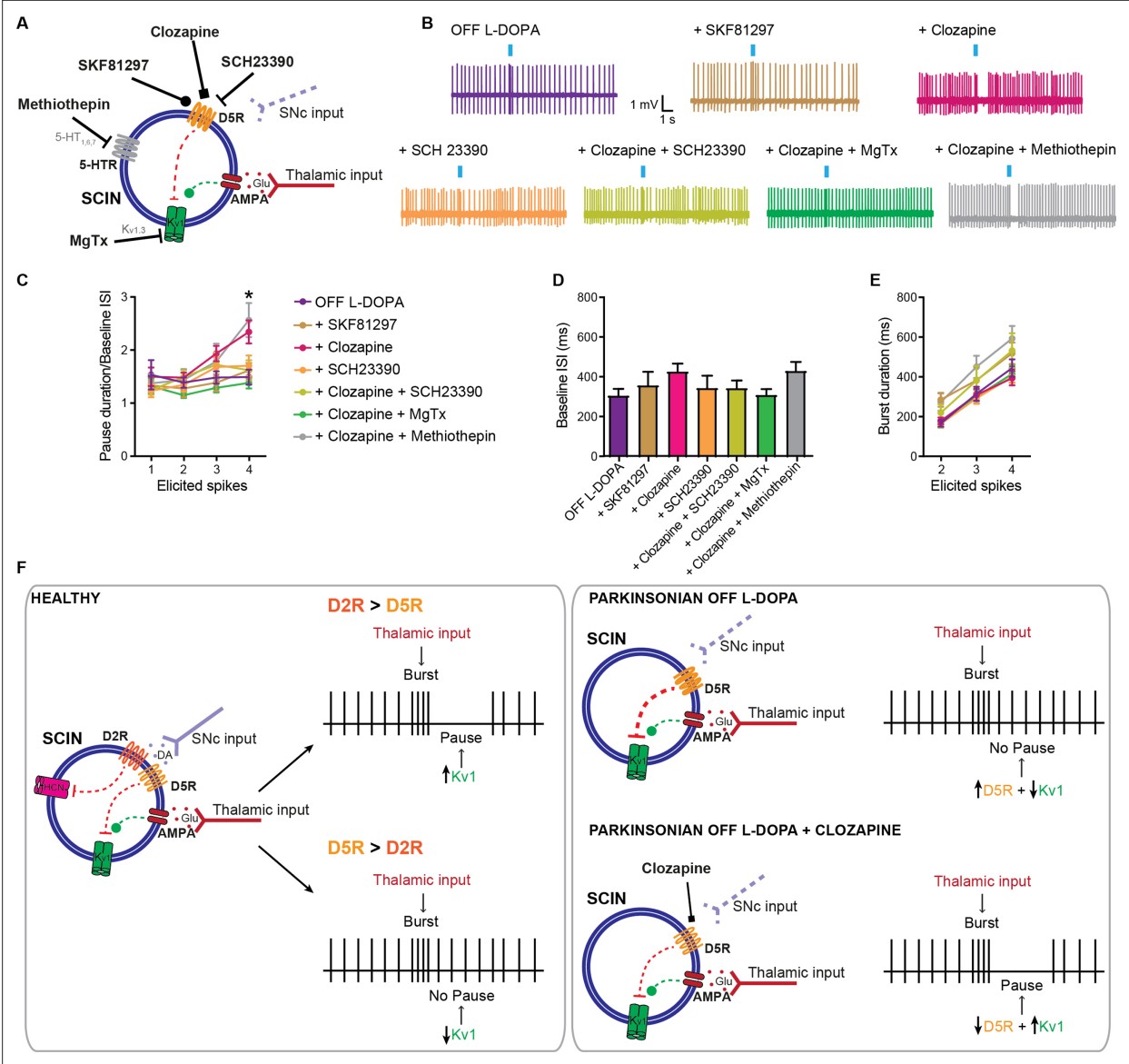

**Figure 7.** The D1/D5 receptor inverse agonist clozapine restores the pause response in striatal cholinergic interneurons (SCINs) from OFF L-DOPA parkinsonian mice. (**A**) Schematic representation of the tested hypothesis. (**B**) Representative responses of SCINs from OFF L-DOPA dyskinetic mice to optogenetic activation of thalamic terminals, with or without SKF81297, clozapine (10 μM; D1/D5 receptor inverse agonist), SCH23390 (10 μM; D1/D5 selective antagonist), clozapine+SCH23390, clozapine+methiothepin (10 μM; 5-HT receptor agonist) or clozapine+MgTx, in the bath. (**C**) Pause duration/baseline interspike interval (ISI) of SCINs from OFF L-DOPA dyskinetic mice that responded with 1, 2, 3, or 4 spikes to optogenetic activation of thalamic terminals (two-way RM ANOVA, interaction *p=0.0083, Dunnett's multiple comparisons test: 4 elicited spikes: OFF L-DOPA vs clozapine: *p=0.0193; OFF L-DOPA vs clozapine+methiothepin: *p=0.0434). (**D**) Baseline ISI of SCINs that responded with 1, 2, 3, or 4 spikes to optogenetic activation of thalamic terminals, under the above conditions (one-way ANOVA, ns). (**E**) Burst duration of SCINs from dyskinetic mice OFF L-DOPA that responded with 2, 3, or 4 spikes to optogenetic activation of thalamic terminals, under the above conditions (two-way RM ANOVA, interaction ns, treatment ns). Mean ± SEM; n=6–17 cells per group, from >5 mice. (**F**) Schematic representation of conclusion. Left, in physiologic conditions, the balance of D2 vs D5 activation emerges as a determinant of the Kv1-dependent pause expression; right, in parkinsonian mice, an imbalance toward D5 signaling abolishes the pause (top), which can be reset with D5 inverse agonism (bottom).

The online version of this article includes the following source data for figure 7:

**Source data 1.** The D1/D5 receptor inverse agonist clozapine restores the pause response in striatal cholinergic interneurons (SCINs) from OFF L-DOPA parkinsonian mice.

by showing that specific blockers of channels constituted by Kv1.3 and Kv1.1 subunits suppress the pause that follows action potential bursts induced by thalamostriatal inputs in adult mice. Although SCINs express inwardly rectifying Kir2 channels capable of amplifying hyperpolarizations (*Wilson, 2005*; *Paz et al., 2021*), our findings suggest they do not contribute to the pause, likely because of insufficient membrane hyperpolarization to recruit Kir currents.

Remarkably, the duration of the SCIN pause response is not only dependent on intrinsic mechanisms but also shaped by thalamic inputs. In vivo, SCIN spontaneous firing synchronizes with excitatory synaptic input fluctuations (*Krok et al., 2023*). The phasic SCIN response may reflect synchronization with thalamic inputs, which themselves exhibit a burst-pause-rebound firing pattern in response to reward-related stimuli in vivo (*Matsumoto et al., 2001*). The potential influence of thalamic rebound activity on pause duration has not been examined before in brain slices. We observed that a second bout of stimulation of thalamic afferents shortens the pause, although this effect is delayed by additional spikes in the initial SCIN burst, potentially recruiting a more robust Kv1 current (*Tubert et al., 2016*).

Importantly, a significant pause could be detected even when only a single spike was evoked by thalamic afferent stimulation. This finding aligns with previous studies showing that SCIN Kv1 currents are recruited by thalamic input trains that elicit subthreshold depolarizing responses, leading to an AHP (*Tubert et al., 2016*). This is particularly interesting given that, in behaving animals, SCIN pauses are not always preceded by a burst of spikes. Significant variability has been observed in the phasic responses of TANs (*Zhang and Cragg, 2017*) and SCINs (*Duhne et al., 2024*) to reward-related stimuli in behaving animals, which appears to depend on multiple factors, including striatal location and whether outcomes are appetitive or aversive. While our ex vivo model may not fully capture the complexity of these responses, it successfully reproduces a key feature: the disappearance of the pause response in TANs following the induction of parkinsonism in monkeys. This provides a valuable opportunity to investigate the underlying mechanisms for the first time.

Recent in vivo studies showed anticorrelated changes of DA and ACh striatal levels, not only during rewarded tasks but also during spontaneous behavior (*Chantranupong et al., 2023*; *Krok et al., 2023*). Several studies support that DA positively modulates the pause response of SCINs through D2R-mediated effects (*Watanabe and Kimura, 1998*; *Ding et al., 2010*; *Kharkwal et al., 2016*), and constitutive ablation of D2Rs from cholinergic neurons disrupts the anticorrelated DA-ACh signal during specific portions of an operant task (*Chantranupong et al., 2023*). However, striatal ACh transients persist and even increase after blockade of striatal DA receptors or lesion of nigrostriatal neurons, while they are reduced after intrastriatal infusion of glutamate receptor antagonists (*Krok et al., 2023*). Although these findings suggest that slow variations of striatal glutamate, rather than DA, control ACh release from SCINs at the space and timescales resolved by fiber photometry (*Krok et al., 2023*), they do not rule out a fine dopaminergic regulation of phasic SCIN responses to glutamatergic inputs, which could be embedded within broader modulations of DA and ACh levels. Our ex vivo data support that endogenous activation of D2R modulates the pause induced by thalamic afferents in SCINs. Moreover, we found that ambient DA in the slice does not modulate SCIN activity through D5Rs, but pharmacological stimulation of D1/D5Rs markedly inhibited the pause response, likely through inhibitory effects of D5Rs on Kv1 channels. Overall, our data show that, in physiological conditions, Kv1 channels are necessary for the expression of the pause response in SCINs of adult mice, and that the pause can be modulated by D5Rs in addition to D2Rs. This D5R modulation may be relevant under conditions of phasic DA release producing synaptic concentrations of DA capable of activating the low-affinity D5Rs (*Figure 6F*).

Learning deficits are believed to play a role in the development of Parkinson's disease symptoms. A recent study demonstrates that practice worsens the performance of previously learned motor skills in parkinsonian mice, while practice combined with therapeutic doses of L-DOPA prevents skill deterioration (*Cheung et al., 2023*). While this learning deficit may be due to the loss of DA, a possible contribution of SCIN dysfunction cannot be dismissed given that, in parkinsonian monkeys, the pause response of presumptive SCINs is lost and can be partially restored with apomorphine (*Aosaki et al., 1994a*).

Our ex vivo studies provide a possible explanation for this pause response loss. We reported that the Kv1 current is diminished in parkinsonian mice and further suppressed 24 hr after the termination of a dyskinetogenic L-DOPA treatment (*Tubert et al., 2016*; *Paz et al., 2022*). Consistent with a key

role of this current in pause generation, we find that the pause that follows a burst of spikes induced by stimulation of thalamostriatal afferents in our ex vivo preparation is markedly diminished in this animal model. Importantly, in previous work, we reported that suppression of the SCIN Kv1 current in dyskinetic mice is caused by sustained cAMP signaling, likely due to enhanced ligand-independent activity of D5Rs, as demonstrated by showing that inverse agonists of D5Rs acutely restore the Kv1 current (*Paz et al., 2022*). In the current study, we find that clozapine, an atypical antipsychotic that behaves as an inverse agonist of D5Rs in the absence of an agonist (*Zhang et al., 2014*), reinstates the pause response of SCINs in OFF L-DOPA dyskinetic mice. Although clozapine also has D1/D5R antagonist properties, these effects would not manifest when DA has been depleted, as is the case in the OFF L-DOPA condition. Consistently, the prototypical D1/D5R antagonist SCH23390, which lacks inverse agonist activity on these receptors (*Zhang et al., 2014*; *Tiberi and Caron, 1994*), neither reinstates the pause response nor alters SCIN spontaneous firing. However, SCH23390 blocks clozapine's effect, supporting the conclusion that clozapine restores the pause response by reducing ligand-independent D5R activity. In addition, the pause restoring action of clozapine cannot be attributed to effects mediated by serotonin receptors as it persisted in the presence of a serotonin receptor antagonist.

Remarkably, while the Kv1.3 blocker MgTx has no effect on SCINs in dyskinetic mice, as expected given the loss of Kv1 currents in this animal model (*Paz et al., 2022*), the pause reinstated by clozapine is suppressed by MgTx. This is consistent with previous findings showing that D5R inverse agonists reinstate the Kv1 current in dyskinetic mice (*Paz et al., 2022*) and supports the key role of the Kv1 current in the SCIN pause and in the dysfunction observed in SCINs after DA neuron lesion and L-DOPA treatment (*Figure 6F*). While our prior work has implicated sustained cAMP signaling in the suppression of the Kv1 current in dyskinetic animals (*Paz et al., 2022*), further work is needed to link cAMP levels to the loss of the SCIN pause.

Importantly, since in DA-intact mice the SCIN pause induced by thalamic inputs is promoted by endogenous stimulation of D2-type receptors (our present findings and *Aosaki et al., 1994a*; *Watanabe and Kimura, 1998*; *Ding et al., 2010*; *Kharkwal et al., 2016*), lack of stimulation of these receptors in DA-depleted mice could be expected to contribute to the absence of the pause in this condition. However, the activation of D2Rs was not sufficient to rescue the pause in dyskinetic mice in our slice preparation. Additionally, D2 agonists failed to modify baseline firing rate in this condition, despite its known inhibitory effects in DA-intact mice (*Maurice et al., 2004*) and partially DA-depleted mice (*McKinley et al., 2019*). Although not addressed in the present study, understanding why D2 agonists fail to restore the SCIN pause is an important question that warrants further investigation. Interestingly, previous studies have reported paradoxical effects of D2 agonists in SCINs in animal models of dystonia (*Pisani et al., 2006*), as well as under conditions where the SCIN's constitutively active integrated stress response is diminished (*Helseth et al., 2021*).

In conclusion, our study demonstrates that Kv1 channels are essential for the pause response of SCINs. We also find that D1/D5R dysregulation of Kv1 channels may contribute to the loss of this pause response in parkinsonian animals chronically treated with L-DOPA. Previously, in the same animal model, we showed that D1/D5 inverse agonists normalize the spontaneous firing pattern and membrane excitability of SCINs by restoring various membrane currents, including Kv1 (*Paz et al., 2022*). While antipsychotics having D5R inverse agonist properties bind to multiple metabotropic receptors that could contribute to their effects, our data highlight a key role of ligand-independent activity of D5Rs in the absence of the pause response in the OFF L-DOPA condition.

## Methods

### Animals

Adult male and female mice (P60-250) allowing the identification of SCINs through cell-type-specific expression of the fluorescent protein tdTomato (Chat^Cre/+;Rosa26^LSL-tdTomato/+, for simplicity) were obtained as previously reported (*Tubert et al., 2016*; *Paz et al., 2021*). Briefly, Chat^Cre/+;Rosa26^LSL-tdTomato/+ mice were Chat^Cre/+ and Rosa26^LSL-tdTomato/+, and were obtained by crossing Chat^Cre/+ (B6;129S6-Chat tm2(cre)Lowl, stock 6410, The Jackson Laboratories) and Rosa-CAG-^LSL-tdTomato-WPRE+/+ mice (B6. CgGt(ROSA)26Sor tm14(CAG-tdTomato)Hze, stock 7914, The Jackson Laboratories). Up to six mice were housed per cage with water and food ad libitum and a 12:12 hr light/dark cycle (lights on at

7:00 AM). Animals were cared for in accordance with institutional regulations (IACUC of the School of Medicine, University of Buenos Aires, 2023-891).

## 6-Hydroxydopamine lesion

Under deep surgical anesthesia (isoflurane, 3% in $O_2$ for induction, 0.5–1% for maintenance), each mouse was mounted in a stereotaxic frame (Kopf Instruments; USA). The neurotoxin 6-hydroxydopamine-HBr (6-OHDA; Sigma-Aldrich AB) was dissolved in 0.1% ascorbate-saline at the concentration 4 µg/µl (free-base). The skull was exposed, and a small hole was drilled at the desired injection site. The following stereotaxic coordinates were used to target the left medial forebrain bundle (MFB): 1 mm posterior (from Bregma), 1.1 mm lateral, and 4.8 mm ventral from dura (*Paxinos and Franklin, 2001*). 1 µl was injected in adult mice (P90–110) at a rate of 0.5 µl/min using a 300 µm diameter cannula attached to a 25 µl Hamilton syringe controlled by a motorized pump (Bioanalytical Systems, USA). The injection cannula was left in place for 6 additional minutes before slowly retracting it. Littermates of lesioned mice received an injection of 0.1% ascorbate-saline in the MFB (sham group). After surgery, each mouse was daily weighed and received a subcutaneous injection of saline and an enriched diet. These cares continued until animals began to regain weight (7–20 days after surgery). To assess behavioral signs of parkinsonism, animals were subjected to extensive behavioral testing 3 weeks after 6-OHDA injection in 3 nonconsecutive days (*Figure 4A*), during the light phase, by an investigator blind to treatment, as described in *Escande et al., 2016*. Briefly, spontaneous ipsilateral rotation in a novel open field (40×40 cm$^2$) is assessed using an automated video-tracking system (Anymaze). A rotation asymmetry index is computed by expressing the number of ipsilateral rotations relative to the total number of rotations. In the 'cylinder test', each mouse is placed in a transparent acrylic cylinder (10 cm diameter and 14 cm high) and videotaped for 5 min. A limb use asymmetry score is computed by expressing the number of wall contacts performed with the forepaw contralateral to the lesion relative to the total number of wall contacts performed with the forepaws. Finally, motor coordination is assessed in an accelerating rotarod (from 4 to 40 rpm in 5 min; Ugo Basile, Italy). The latency to fall from the rod is automatically recorded with a cutoff time of 5 min. Each animal was assessed five times with 5 min intertrial intervals in a single day.

## L-DOPA treatment

Mice were treated with 12 mg/kg of L-3,4-dihydroxyphenylalanine methyl ester hydrochloride (L-DOPA, Sigma-Aldrich) combined with 12 mg/kg of benserazide (Sigma-Aldrich). L-DOPA and benserazide were diluted in 0.9% saline solution. AIMs were rated as previously described (*Keifman et al., 2019*). Briefly, AIMs rating started after 20 min of L-DOPA injection, and mice were rated for 1 min every 20 min during 2 hr. Each subtype of dyskinesia (oral, axial, and forelimb) was scored on a scale ranging from 0 to 4 (where 0=no dyskinesia; 1=occasional dyskinesia displayed for <50% of the observation time; 2=sustained dyskinesia displayed for >50% of the observation time; 3=continuous dyskinesia; 4=continuous dyskinesia not interruptible by external stimuli).

## ChR2-EYFP adeno-associated vector injection

Anesthesia and surgical procedures were as indicated for the nigrostriatal lesions. A CaMKII-driven ChR2-EYFP expressing adeno-associated virus (rAAV2/CaMKII-hChR2(H134R)-EYFP-WPRE-PA) was injected in the thalamus at the following coordinates: –1.3 mm posterior from Bregma, 0.8 mm lateral, and –3 mm ventral from dura (*Paxinos and Franklin, 2001*). 0.5 µl of viral vector was delivered using a glass micropipette (World Precision Instruments) pulled with a vertical glass puller (Narishige). Experiments were performed after at least 28 postoperative days (*Figure 1A and B*).

## Patch clamp recordings

Twenty-four hours after the last L-DOPA shot, mice were anesthetized with isoflurane and decapitated for brain slicing. The brain was quickly removed, chilled in ice-cold low-$Ca^{2+}$/high-$Mg^{2+}$ artificial cerebrospinal fluid (ACSF), and prepared for slicing. Coronal slices (300 µm) at the level of the striatum were cut with a vibratome (Pelco T series 1000, Ted Pella) and were incubated in low-$Ca^{2+}$/high-$Mg^{2+}$ ACSF at 34°C for 30 min and then at room temperature. ACSF composition was as follows (in mM): 125 NaCl, 2.5 KCl, 1.3 $NaH_2PO_4·H_2O$, 26 $NaHCO_3$, 2 $CaCl_2$, 1 $MgCl_2$, and 10 glucose. For the low-$Ca^{2+}$/high-$Mg^{2+}$ ACSF, 0.5 mM $CaCl_2$ and 2.5 mM $MgCl_2$ were used. Slices

were transferred to a submersion-type chamber perfused by a peristaltic pump (Ismatec, Germany) with ACSF at a constant rate of 3 ml/min; temperature in the recording chamber was set at 32°C with a TC-344B temperature controller (Warner Instruments). Cells were visualized using an upright microscope (Nikon Eclipse) equipped with a ×40 water-immersion objective, DIC and fluorescence optics, and an infrared camera connected to a monitor and computer. Recording electrodes were made with borosilicate glass capillaries shaped with a puller (P-97, Sutter Instruments). Dorsolateral SCINs were recorded in cell-attached configuration with patch pipettes filled with intracellular solution (in mM): 20 KCl, 120 K-gluconate, 10 HEPES, 3 $Na_2ATP$, 0.3 NaGTP, 0.1 EGTA, 10 phosphocreatine, and 2 $MgCl_2$, pH 7.3 adjusted with KOH. Recordings were amplified (Axopatch-1D; Molecular Devices), sampled at 20 kHz (Digidata 1322A, Molecular Devices) and acquired on a PC running pClamp 9.2 (Molecular Devices). In all the experiments, PIC (100 µM) was always present in the ACSF to block GABAergic transmission. Optogenetic stimulation was generated with a 447 nm light-emitting diode (Tolket, Argentina) and delivered through an optic fiber placed <300 µm from the recorded cell. One to 20, 2-ms-long pulses, at 20 Hz and 10 mW, were delivered 5 s after the initiation of the sweep, for 10 consecutive sweeps (*Figure 1C*). For experiments with a second pulse train, the second train was delivered 350 ms after the end of the first train. The second train contained 1–3 pulses (*Figure 1I*).

## Pharmacological manipulations

Unless otherwise stated, reagents were purchased from Sigma (Argentina). Salts were purchased from Baker (Research AG, Argentina). Drugs were prepared as stock solutions, diluted in ACSF immediately before use, and applied through the perfusion system. The following stock solvents and final concentrations were used: distilled $H_2O$ for BaCl (10 µM), ZD7288 (30 µM), mecamylamine (10 µM, RBI), methiothepin (10 µM), quinpirole (1 µM, 10 µM), and sumanirole (10 µM); DMSO for CNQX (40 µM, Tocris), PIC (100 µM), SKF81297 (2 µM), SCH23390 (10 µM), sulpiride (10 µM, Santa Cruz Biotechnology), XE991 (10 µM), and clozapine (10 µM, Rospaw Laboratory); and the manufacturer's recommended storage buffer (0.1% BSA, 100 mM NaCl, 10 mM Tris pH 7.5, 1 mM EDTA) for MgTx (3 nM and 30 nM, Alomone Labs) and g-DTx (100 nM, Alomone Labs).

## Acquisition and analysis of electrophysiological data

Data acquisition and analysis were performed with ClampFit (Molecular Devices, RRID:SCR_011323). GraphPad Prism version 8.00 for Windows (GraphPad Software, RRID:SCR_002798) and custom-made MATLAB (Mathworks, RRID:SCR_001622) routines were used for data analysis.

For each cell, three to five recordings were made, varying the number of light pulses delivered during the stimulation train. Each recording consisted of 10 sweeps with the same stimulus. Then, the recordings were analyzed by merging all the sweeps where the stimulation elicited the same number of spikes, regardless of the stimulation protocol. Only sweeps that triggered 1–4 spikes in response to the light were included in the analysis. The number of spikes in the burst was determined by the number of spikes fired during the length of the stimulus plus 250 ms after the end of the stimulus. The pause was calculated as the time from the last spike of the burst triggered by the thalamostriatal input activation to the following spike. The average baseline ISI was calculated, taking into account all ISI collected during the 5 s preceding each stimulus. The burst duration was calculated as the time from the first to the last spike in the burst. To determine the influence of spontaneous spikes on evoked spiking, we plotted the time from the stimulus onset to the first spike in the burst (x2, for stimulations that evoked 1 spike) as a function of the time from the spike preceding the stimulus to the stimulus onset (x1). Each value was normalized to the average baseline ISI. For sham stimulation, we performed the same analysis at a random time point during baseline, for the same sweeps used to analyze light stimulus responses.

Each cell was recorded with only one pharmacological treatment. Most animals contributed with at least two neurons that received different drug treatments. For each experiment, control neurons were obtained from the same animals that contributed treated cells, plus a subset of cells from animals in which no cells were treated.

All the data tables are included and available as source data. MATLAB's custom routine is uploaded to GitHub (copy archived at *Paz, 2025*).

## Perfusion

For confirmation of thalamostriatal projections to the striatum, Chat$^{Cre/+}$;Rosa26$^{LSL-tdTomato/+}$ mice were anesthetized with pentobarbital (100 mg/kg, i.p.), perfused through the ascending aorta with saline solution supplemented with heparin (2 U/ml, Rivero, Argentina) followed by 4% paraformaldehyde in 0.1 M phosphate buffer saline 0.1 M, pH 7.4 (PBS). Brains were removed, post-fixed for 24 hr in the same fixative, cryoprotected in 30% sucrose/PBS, and cut. Sections containing the striatum and the thalamus were preserved in PBS with 0.1% sodium azide.

## Immunohistochemistry

To confirm nigral dopaminergic lesions and the site of the AAV injection into the thalamus in mice used for acute slicing, tissue blocks containing thalamus and substantia nigra obtained after striatal slicing were fixed overnight in 4% paraformaldehyde in PBS, cryoprotected in 30% sucrose/PBS and cut at 30 μm with a sliding-freezing microtome (Leica). Sections containing the SNc were preserved in PBS with 0.1% sodium azide. For TH-immunoreactivity, sections were blocked for 2 hr at room temperature (PBS containing 5% NGS and 0.3% Triton X-100), incubated overnight with primary antibodies (mouse anti-TH, Chemicon, 1:1000, RRID:AB_2201528) at 4°C, washed three times in 0.15% Triton X-100/PBS (PBS-T) and incubated with secondary antibodies conjugated to Alexa Fluor488 (Goat anti-mouse IgG, Invitrogen, 1:500, RRID:AB_138404) for 2 hr at room temperature. Sections were then washed three times in PBS-T and one time in PBS, mounted on glass slides with Vectashield (Vector) and cover-slipped. For confirmation of the site of AAV injection, sections containing the thalamus were washed in PBS, mounted on glass slides with Vectashield (Vector) and coverslipped.

## Image acquisition and analysis

Images of mounted sections were acquired with a Zeiss AxioImager M2 microscope with Neurolu-cida software (MFB Bioscience), with ×10, ×20, or ×40 objectives. Images were not modified after acquisition.

## Statistical analyses

Data were tested for normality and homoscedasticity with the Anderson-Darling test and Spearman's rank correlation test, respectively. Nonparametric tests were used when these criteria were not met. Statistical analysis details can be found in the figure legends and wherever numerical data is presented in the text. Significance level was set at $p < 0.05$, and all data are expressed as mean ± SEM unless otherwise specified. GraphPad Prism and MATLAB routines were used for data analysis. Sidak or Dunn's corrections were applied for multiple comparisons.

## Acknowledgements

The authors thank Graciela Ortega, Analía López Díaz, Agostina Presta, Verónica Risso, Lucía Garbini, and Micaela Buscema for expert technical assistance in animal genotyping, histology, and animal care, and Juan Belforte for helpful discussions. We also thank the Rospaw Laboratory for their kind donation of clozapine and Lidia Szczupak for the methiothepin donation. This study was supported by Fondo para la Investigación Científica y Tecnológica (FONCYT; Proyecto de Investigación Científica y Tecnológica [PICT] 2018–2738, 2019–2416, 2020–325, and 2022–42) and Universidad de Buenos Aires (UBACYT2018-305; PIDAE 2025-7).

## Additional information

### Funding

| Funder | Grant reference number | Author |
| --- | --- | --- |
| Fondo para la Investigación Científica y Tecnológica | PICT 2020-325 | Mario Gustavo Murer |

| Funder | Grant reference number | Author |
| --- | --- | --- |
| Universidad de Buenos Aires | PIDAE 2025-7 | Mario Gustavo Murer |
| Universidad de Buenos Aires | UBACYT2018-305 | Mario Gustavo Murer |

The funders had no role in study design, data collection and interpretation, or the decision to submit the work for publication.

## Author contributions

Cecilia Tubert, Rodrigo Manuel Paz, Conceptualization, Data curation, Formal analysis, Funding acquisition, Investigation, Methodology, Writing - original draft, Writing - review and editing; Agostina Mónica Stahl, Kianny Miroslava Sanchez Armijos, Data curation; Lorena Rela, Conceptualization, Supervision, Methodology, Writing - review and editing; Mario Gustavo Murer, Conceptualization, Supervision, Funding acquisition, Investigation, Visualization, Methodology, Writing - original draft, Project administration, Writing - review and editing

## Author ORCIDs

Cecilia Tubert ⓘ https://orcid.org/0000-0001-6766-5849
Rodrigo Manuel Paz ⓘ http://orcid.org/0000-0002-5179-2760
Kianny Miroslava Sanchez Armijos ⓘ https://orcid.org/0000-0003-0164-7319
Mario Gustavo Murer ⓘ https://orcid.org/0000-0001-5149-1311

## Ethics

This study was performed in strict accordance with the recommendations in the Guide for the Care and Use of Laboratory Animals (CICUAL) of the University of Buenos Aires Scoold of Medicine. All of the animals were handled according to approved institutional animal care and use committee (CICUAL) protocols of the University of Buenos Aires (Resolution RESCD-2021-2619-E-UBA-DCT#FMED). All surgery was performed under isofluorane anesthesia, and every effort was made to minimize suffering.

Reviewer #1 (Public review): https://doi.org/10.7554/eLife.102184.4.sa1
Reviewer #2 (Public review): https://doi.org/10.7554/eLife.102184.4.sa2
Reviewer #3 (Public review): https://doi.org/10.7554/eLife.102184.4.sa3
Author response https://doi.org/10.7554/eLife.102184.4.sa4

# Additional files

## Supplementary files
MDAR checklist

## Data availability
All data generated or analysed during this study are included in the manuscript.

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
