## [Editor Report · eLife Assessment]

The authors investigated the mechanisms underlying the pause in striatal cholinergic interneurons (SCINs) induced by thalamic input, identifying that Kv1 channels play a key role in this burst-dependent pause. The experimental evidence is **convincing**.

The study provides **important** mechanistic insights into how burst activity in SCINs leads to a subsequent pause, highlighting the involvement of D1/D5 receptors.

---

## [Referee Report · Reviewer #1 (Public review)]

Summary:

Tubert C. et al. investigated the role of dopamine D5 receptors (D5R) and their downstream potassium channel, Kv1, in the striatal cholinergic neuron pause response induced by thalamic excitatory input. Using slice electrophysiological analysis combined with pharmacological approaches, the authors tested which receptors and channels contribute to the cholinergic interneuron pause response in both control and dyskinetic mice (in the L-DOPA off state). They found that activation of Kv1 was necessary for the pause response, while activation of D5R blocked the pause response in control mice. Furthermore, in the L-DOPA off state of dyskinetic mice, the absence of the pause response was restored by the application of clozapine. The authors claimed that (1) the D5R-Kv1 pathway contributes to the cholinergic interneuron pause response in a phasic dopamine concentration-dependent manner, and (2) clozapine inhibits D5R in the L-DOPA off state, which restores the pause response.

Strengths:

The electrophysiological and pharmacological approaches used in this study are powerful tools for testing channel properties and functions. The authors' group has well-established these methodologies and analysis pipelines. Indeed, the data presented were robust and reliable.

The authors addressed all concerns I raised. Presented data are convincing and support their claims.

---

## [Referee Report · Reviewer #2 (Public review)]

Summary:

This manuscript by Tubert et al. presents the role of D5 receptors (D5R) in regulating the striatal cholinergic interneuron (CIN) pause response through D5R-cAMP-Kv1 inhibitory signaling. Their findings provide a compelling model explaining the "on/off" switch of the CIN pause, driven by the distinct dopamine affinities and the balance of D2R and D5R. Furthermore, the study bridges their previous finding of CIN hyperexcitability (Paz et al., Movement Disorder 2022) with the loss of the pause response in LID mice and demonstrates the restore of the pause through D1/D5 inverse agonist clozapine.

Strengths:

The study presents solid findings, and the writing is logically structured and easy to follow. The experiments are well-designed, properly combining ex vivo electrophysiology recording, optogenetics, and pharmacological treatment to dissect / rule out most, if not all, alternative mechanisms in their model.

Weaknesses (fixed in this revision):

In this round of revision, the authors have included additional experiments examining the role of D2R, and the possible clozapine effects on serotonin receptors in the LID off -L-DOPA ex vivo slices. Although, to our surprise, D2R agonism using quinpirole and sumanirole failed to restore the CIN pause, this study still provides new insights into the balance between D2R and D5R in modulating CIN pause.

Overall, the authors' response adequately addressed concerns raised in the previous revision.

---

## [Author Response]

The following is the authors’ response to the previous reviews

**Public Reviews:**

**Reviewer #1 (Public review):**
Summary:Tubert C. et al. investigated the role of dopamine D5 receptors (D5R) and their downstream potassium channel, Kv1, in the striatal cholinergic neuron pause response induced by thalamic excitatory input. Using slice electrophysiological analysis combined with pharmacological approaches, the authors tested which receptors and channels contribute to the cholinergic interneuron pause response in both control and dyskinetic mice (in the L-DOPA off state). They found that activation of Kv1 was necessary for the pause response, while activation of D5R blocked the pause response in control mice. Furthermore, in the L-DOPA off state of dyskinetic mice, the absence of the pause response was restored by the application of clozapine. The authors claimed that (1) the D5R-Kv1 pathway contributes to the cholinergic interneuron pause response in a phasic dopamine concentration-dependent manner, and (2) clozapine inhibits D5R in the L-DOPA off state, which restores the pause response.StrengthsThe electrophysiological and pharmacological approaches used in this study are powerful tools for testing channel properties and functions. The authors' group has well-established these methodologies and analysis pipelines. Indeed, the data presented were robust and reliable.Weaknesses:Although the paper has strengths in its methodological approaches, there is a significant gap between the presented data and the authors' claims.The authors answered the most of concerns I raised. However, the critical issue remains unresolved.I am still not convinced by the results presented in Fig. 6 and their interpretation. Since Clozapine acts as an agonist in the absence of an endogenous agonist, it may stimulate the D5R-cAMP-Kv1 pathway. Stimulation of this pathway should abolish the pause response mediated by thalamic stimulation in SCINs, rather than restoring the pause response. Clarification is needed regarding how Clozapine reduces D5R-ligand-independent activity in the absence of dopamine (the endogenous agonist). In addition, the author's argued that D5R antagonist does not work in the absence of dopamine, therefore solely D5R antagonist didn't restore the pause response. However, if D5R-cAMP-Kv1 pathway is already active in L-DOPA off state, why D5R antagonist didn't contribute to inhibition of D5R pathway? Since Clozapine is not D5 specific and Clozapine experiments were not concrete, I recommend testing whether other receptors, such as the D2 receptor, contribute to the Clozapine-induced pause response in the L-DOPA-off state.

Thank you for the opportunity to clarify this point. It seems there may have been a misunderstanding regarding our proposal about clozapine's mechanism of action. We are not suggesting that clozapine acts as an agonist, but rather as an “inverse agonist”. Unlike classical agonists, inverse agonists produce a pharmacological effect opposite to that of an agonist. Although clozapine is best known for its antagonistic effects on dopamine and serotonin receptors, under conditions where no endogenous agonist is present, it has been shown to reduce the constitutive activity of D1 and D5 receptors (PMID: 24931197). This is explained in lines 240-254 in the Results section.

In contrast, the prototypical and selective D1/D5 receptor antagonist SCH23390 does not exhibit inverse agonist properties and would not be expected to produce effects in the absence of an agonist (PMID: 7525564). The observation that SCH23390 blocks the effects of clozapine in dopamine-depleted animals strongly supports the idea that clozapine acts through D1/D5 receptors. This is now clarified in lines 257264.

To further address your comments, we now include a new figure (Figure 6) presenting experiments that show D2-type receptor agonists do not restore the pause response in dyskinetic mice in the off-L-DOPA condition. These results are described in a new subsection of the Results section and discussed in a newly added paragraph in the Discussion (lines 369-380).

Finally, to exclude a potential contribution of serotonin receptors to clozapine’s effects, we have expanded what is now Figure 7 (formerly Figure 6) to show that clozapine continues to restore the pause response even in the presence of a serotonin receptor antagonist in the bath.

All these results are further discussed in lines 342-360.

**Reviewer #2 (Public review):**
Summary:This manuscript by Tubert et al. presents the role of D5 receptors (D5R) in regulating the striatal cholinergic interneuron (CIN) pause response through D5R-cAMP-Kv1 inhibitory signaling. Their findings provide a compelling model explaining the "on/off" switch of the CIN pause, driven by the distinct dopamine affinities of D2R and D5R. This mechanism, coupled with varying dopamine states, is likely critical for modulating synaptic plasticity in cortico-striatal circuits during motor learning and execution. Furthermore, the study bridges their previous finding of CIN hyperexcitability (Paz et al., Movement Disorder 2022) with the loss of the pause response in LID mice and demonstrates the restore of the pause through D1/D5 inverse agonism.Strengths:The study presents solid findings, and the writing is logically structured and easy to follow. The experiments are well-designed, properly combining ex vivo electrophysiology recording, optogenetics, and pharmacological treatment to dissect / rule out most, if not all, alternative mechanisms in their model.Weaknesses:While the manuscript is overall satisfying, one conceptual gap needs to be further addressed or discussed: the potential "imbalance" between D2R and D5R signaling due to the ligand-independent activity of D5R in LID. Given that D2R and D5R oppositely regulate CIN pause responses through cAMP signaling, investigating the role of D2R under LID off L-DOPA (e.g., by applying D2 agonists or antagonists, even together with intracellular cAMP analogs or inhibitors) could provide critical insights. Addressing this aspect would strengthen the manuscript in understanding CIN pause loss under pathological conditions.

Thank you for your comments. Although our primary focus is on the role of D5 receptors, we have also investigated the effects of two D2-type receptor agonists in dyskinetic mice in the off-L-DOPA condition. We found that neither quinpirole nor sumanirole was able to restore the pause response. These results are presented in Figure 6 and related text in the Results and Discussion sections.

Understanding why D2 agonists fail to restore the pause response—despite their expected effect of reducing cAMP levels—is an important question that warrants further investigation. Interestingly, previous studies have reported paradoxical effects of D2 receptor stimulation in SCINs in animal models of dystonia (PMID: 16934985, PMID: 21912682), as well as under conditions where the SCIN’s constitutively active integrated stress response is diminished (PMID: 33888613). This is now discussed in lines 369-380.

**Reviewer #3 (Public review):**
Summary:Tubert et al. investigate the mechanisms underlying the pause response in striatal cholinergic interneurons (SCINs). The authors demonstrate that optogenetic activation of thalamic axons in the striatum induces burst activity in SCINs, followed by a brief pause in firing. They show that the duration of this pause correlates with the number of elicited action potentials, suggesting a burst-dependent pause mechanism. The authors demonstrated this burst-dependent pause relied on Kv1 channels. The pause is blocked by a SKF81297 and partially by sulpiride and mecamylamine, implicating D1/D5 receptor involvement. The study also shows that the ZD7288 does not reduce the duration of the pause, and that lesioning dopamine neurons abolishes this response, which can be restored by clozapine.Weaknesses:While this study presents an interesting mechanism for SCIN pausing after burst activity, there are several major concerns that should be addressed:(1) Scope of the Mechanism: It is important to clarify that the proposed mechanism may apply specifically to the pause in SCINs following burst activity. The manuscript does not provide clear evidence that this mechanism contributes to the pause response observed in behavioral animals. While the thalamus is crucial for SCIN pauses in behavioral contexts, the exact mechanism remains unclear. Activating thalamic input triggers burst activity in SCINs, leading to a subsequent pause, but this mechanism may not be generalizable across different scenarios. For instance, approximately half of TANs do not exhibit initial excitation but still pause during behavior, suggesting that the burstdependent pause mechanism is unlikely to explain this phenomenon. Furthermore, in behavioral animals, the duration of the pause seems consistent, whereas the proposed mechanism suggests it depends on the prior burst, which is not aligned with in vivo observations. Additionally, many in vivo recordings show that the pause response is a reduction in firing rate, not complete silence, which the mechanism described here does not explain. Please address these in the manuscript.

Thank you for the opportunity to clarify these points. We acknowledge that the response of SCINs to optogenetic stimulation of thalamic afferents in brain slices represents a model system that may not capture all aspects of TAN responses to behaviorally salient events. Nevertheless, this approach allows us to test mechanistic hypotheses that are difficult to address in behaving animals with current technologies. This is now stated in lines 311-314.

Importantly, our ex vivo preparation reproduces, for the first time, the loss of TAN responses observed in non-human primates with parkinsonism, enabling investigation of the underlying mechanisms. In line with your suggestion, we have expanded the Discussion (third and fourth paragraphs) to address the sources of variability in pause responses.

(2) Terminology: The use of "pause response" throughout the manuscript is misleading. The pause induced by thalamic input in brain slices is distinct from the pause observed in behavioral animals. Given the lack of a clear link between these two phenomena in the manuscript, it is essential to use more precise terminology throughout, including in the title, bullet points, and body of the manuscript.

Thank you for raising this important point. We agree that it is essential to be precise in describing the nature of the pause observed in our ex vivo model. While we believe that readers would recognize from the abstract and methods that our study focuses on a model of the pause response, we understand your concern about potential confusion. In response, we have revised the terminology in the abstract, bullet points, and throughout the manuscript to more clearly reflect that we are describing an ex vivo model of the pause observed in behaving animals.

(3) Kv1 Blocker Specificity: It is unclear how the authors ruled out the possibility that the Kv1 blocker did not act directly on SCINs. Could there be an indirect effect contributing to the burst-dependent pause?

Clarification on this point would strengthen the interpretation of the results.

This issue is addressed in lines 147-150.

(4) Role of D1 Receptors: While it is well-established that activating thalamic input to SCINs triggers dopamine release, contributing to SCIN pausing (as shown in Figure 3), it would be helpful to assess the extent to which D1 receptors contribute to this burst-dependent pause. This could be achieved by applying the D1 agonist SKF81297 after blocking nAChRs and D2 receptors.

Figure 3C shows that the D1/D5 receptor antagonist SCH23390 does not modify the pause, while the full D1/D5 agonist SKF81297 abolishes it, indicating that in our slice preparation, baseline dopamine levels are not contributing to the pause through D1/D5 receptor stimulation.

(5) Clozapine's Mechanism of Action: The restoration of the burst-dependent pause by clozapine following dopamine neuron lesioning is interesting, but clozapine acts on multiple receptors beyond D1 and D5. Although it may be challenging to find a specific D5 antagonist or inverse agonist, it would be more accurate to state that clozapine restores the burst-dependent pause without conclusively attributing this effect to D5 receptors.

As explained in our response to Reviewer #1, the effect of clozapine is blocked by the D1/D5-selective antagonist SCH23390. Additionally, new data presented in Figure 7C show that clozapine's ability to restore the pause response is maintained even in the presence of a broad-spectrum serotonin receptor antagonist. Since SCINs do not significantly express D1 receptors, we believe that these findings strongly support a role for D5 receptors in SCINs.

Comments on revisions:The authors have addressed many of my concerns. However, I remain unconvinced that adding an 'ex vivo' experiment fully resolves the fundamental differences between the burst-dependent pause observed in slices - defined by the duration of a single AHP - and the pause response in CHINs observed in vivo, which may involve contributions from more than one prolonged AHP. In vivo, neurons can still fire action potentials during the pause, albeit at a lower frequency. Moreover, in behaving animals, pause duration does not vary with or without initial excitation. The mechanism proposed demonstrates that the pause duration, defined by the length of a single AHP, is positively correlated with preceding burst activity.

As discussed in paragraphs 3 and 4 of the Discussion (starting at line 285), and illustrated in Figure 1J–K, our data show that the duration of the pause can be modulated by rebound excitation from thalamic input. The absence of this rebound allows us to observe a longer pause when more spikes are elicited during the initial excitatory phase, providing a clearer readout of the contribution of intrinsic membrane mechanisms. We do not claim that intrinsic mechanisms alone account for the entire phasic response of SCINs in behaving animals (lines 295-303 in Discussion).

To improve clarity, I recommend using the term 'SCIN pause' to describe the ex vivo findings, distinguishing them more explicitly from the 'pause response' observed in behaving animals. This distinction would help contextualize the ex vivo findings as potentially contributing to, but not fully representing, the pause response in vivo.

We did changes in the abstract, bullet points, and main text to clarify that we are not studying the in vivo response.

Again, it would be helpful to present raw data for pause durations rather than relying solely on ratios. This approach would provide the audience with a clearer understanding of the absolute duration of the burst-dependent pause and allow for better comparison to the ~200 ms pause observed in behaving animals.

Thank you for your comment. Following your suggestion, we provide the average pause durations for the data shown in Figure 1H (lines 127–130). We opted not to include raw pause durations in the main text for all figures, as this would make the manuscript more difficult to read and, in our view, is unnecessary. The figures already allow readers to estimate absolute durations: in each case, pause durations are shown relative to baseline ISIs in one panel, while the corresponding absolute ISIs are shown side-by-side. This provides a clearer understanding of pause magnitude relative to the cell’s spontaneous firing, which is more informative than absolute values alone, since one would expect a pause to be longer than the average ISI. Please note that baseline ISI are significantly shorter in dyskinetic mice (Figure 5l). Showing the pause duration relative to baseline ISI allows readers to readily compare results across figures regardless of changes in SCIN baseline firing rate.

Additionally, it is important to note that, in vivo, pause durations are typically inferred from perievent time histograms (PETHs), which represent population averages across many trials. In contrast, in our ex vivo studies, we measured pause duration on a trial-by-trial basis. This approach enables us to analyze how the pause duration varies as a function of the initial burst size in the same neuron—something not typically reported in in vivo studies. As described in the first two paragraphs of the Results, the same SCIN may respond with a different number of spikes in successive trials, and this variability is influenced by factors such as the timing of the last spontaneous spike relative to stimulation onset (Figure 1D–F). We are not aware of studies reporting trial-by-trial analyses of pause duration in behaving animals, particularly in relation to the strength of initial excitation. Therefore, while our slice preparation may yield pause durations that are longer than those observed in vivo, direct comparison to PETH-derived pause durations from behaving animals is not straightforward.